# A Case Report of Immunotherapy-Resistant MSI-H Gastric Cancer with Significant Intrapatient Tumoral Heterogeneity Characterized by Histologic Dedifferentiation

**DOI:** 10.3390/jcm11123413

**Published:** 2022-06-14

**Authors:** Nikhila Kethireddy, Leonidas Arvanitis, Janine LoBello, Yanghee Woo, Szabolcs Szelinger, Joseph Chao

**Affiliations:** 1Department of Hematology/Oncology, Harbor-UCLA Medical Center, Torrance, CA 90502, USA; nkethireddy@dhs.lacounty.gov; 2Department of Pathology, City of Hope Comprehensive Cancer Center, Duarte, CA 91010, USA; larvanitis@coh.org; 3Exact Sciences, Phoenix, AZ 85004, USA; jlobello@exactsciences.com (J.L.); sszelinger@exactsciences.com (S.S.); 4Division of Surgical Oncology, Department of Surgery, City of Hope Comprehensive Cancer Center, Duarte, CA 91010, USA; yhwoo@coh.org; 5Department of Medical Oncology and Therapeutics Research, City of Hope Comprehensive Cancer Center, 1500 E. Duarte Road, Duarte, CA 91010, USA

**Keywords:** immunotherapy, microsatellite instability, intratumoral heterogeneity, whole-exome sequencing

## Abstract

We describe a patient with both gastric adenocarcinoma and metastatic squamous cell carcinoma (SCC) of unknown primary site. The possibility of a single malignant clonal process as opposed to differing primaries was supported by the finding of both histologies exhibiting high microsatellite instability. Despite evidence of tumor microsatellite instability, the patient’s disease process did not respond to immune checkpoint inhibition. Our pursuit of whole-exome sequencing and comparing the single-nucleotide variant profiles of both tumors supported a single clonal process with the development of significant intratumoral heterogeneity. High intratumoral heterogeneity has posed a challenge to precision medicine approaches, but we also provide a review of the literature of this phenomenon mediating resistance to immunotherapy strategies.

## 1. Introduction

Gastric cancer (GC) accounts for over 1 million new cases and an estimated 769,000 deaths every year [1]. Distinguishing metastatic recurrence and the development of a second primary carries prognostic and therapeutic implications [2]. Conventionally, the histopathological comparison of tissue specimens has been utilized. However, this has limitations, including interobserver variability, tumor heterogeneity, and changes in the tumor microenvironment that can impact diagnosis [2]. The use of comprehensive genomic profiling (CGP) can augment histopathologic analysis while concurrently pinpointing genomic alterations that have therapeutic potential [2,3,4,5,6]. Clinical trials for targeted therapies in individuals with metastatic GC who were positive for ERBB2, MET, EGFR, and FGFR2 showed variable results [7,8,9,10,11,12,13,14]. One potential reason these trials did not show a significant benefit is that the primary tumor was used to guide treatment [7].

We describe a patient with both gastric adenocarcinoma and squamous cell carcinoma (SCC) of unknown primary site. We postulate that despite different histopathological diagnoses our patient likely had a single metastatic disease process rather than two separate primaries. Our conclusion is supported by comparisons of the single-nucleotide variant (SNV) profiles derived from whole-exome sequencing between the primary and metastatic sites aligning with a single clonal process, demonstrating interlesional heterogeneity. We believe that dedifferentiation from GC to SCC is reflective of substantial intratumoral heterogeneity (ITH) in our patient, which likely led to immune checkpoint inhibitor (ICI) resistance. Written informed consent was obtained from the patient for the collection, analysis, and publication of deidentified molecular and clinical data under an institutional-review-board-approved research protocol.

## 2. Case Report

A 56-year-old Asian female with no past medical history self-palpated a left neck lump. She was briefly residing outside the US and while there was sent for positron emission tomography/computerized tomography (PET/CT), which demonstrated enlarged lymph nodes in the neck, mediastinum, axilla, left gastric region, common hepatic, and retroperitoneum. The scan also revealed irregular wall thickening of the gastric lower body with extramural infiltration. The patient underwent a core biopsy of the left level IV neck lymph node (LN), which showed poorly differentiated carcinoma, favoring metastatic SCC. She had bilateral screening mammograms and breast ultrasounds, which were both unremarkable. Esophagogastroduodenoscopy (EGD) identified an ulcerating infiltrative lesion in the midbody of the stomach. The pathology revealed moderate to poorly differentiated adenocarcinoma. A colonoscopy found benign polyps. 

She returned to the United States and presented to our institution. Her physical exam was significant for mild epigastric tenderness. The lab work, including complete blood counts, carcinoembryonic antigen, CA 19-9, CA 125, CA 15-3, and alpha fetoprotein, was normal. A CT of the abdomen and pelvis demonstrated a mass-like gastric wall thickening, extensive conglomerated lymphadenopathy in the upper abdomen, and peritoneal carcinomatosis with small volume malignant ascites. She was evaluated with a cervical exam and Pap smear, which were normal, ruling out a cervical SCC as a primary origin of the metastatic SCC. Her initial biopsy samples were unobtainable from overseas. A repeat biopsy of a different axillary enlarged LN was pursued. An EGD for biomarker analyses was performed prior to starting any therapy and confirmed a necrotic mass in the body of the stomach. The pathology was consistent with moderately differentiated adenocarcinoma with mucinous features. Biomarker analyses, including programmed death-ligand 1 (PD-L1) expression, showed a combined positive score (CPS) of 0. The HER2/neu immunohistochemistry (IHC) was negative. She had a loss of the nuclear expression of DNA mismatch repair (MMR) protein PMS2 with a partial loss of MLH1 and retained the nuclear expression of MSH2 and MSH6. The biopsy of the axillary LN demonstrated a metastatic SCC. The PD-L1 tumor proportion score (TPS) was 5%, the PD-L1 combined positive score (CPS) was also 5, and the MMR IHC showed a loss of PMS2 and retained nuclear expression of MLH1, MLH2, and MSH6. Both the axillary LN and gastric mass showed a loss of MMR protein PMS2, demonstrating that both tumors had microsatellite instability (MSI-H/dMMR). With the apparent separate tumor histologies all being MSI-H, germline testing for Lynch syndrome was conducted, with no germline pathogenic mutations being detected in *MLH1*, *MSH2*, *MSH6*, *PMS2*, or *EPCAM*. 

FOLFOX and pembrolizumab were initiated, given that the tumors were MSI-H. Whole-exome sequencing of both the adenocarcinoma and the SCC using the Oncomap^TM^ ExTra test confirmed the MSI-H status and numerous somatic alterations (Table 1). The OncomapTM ExTra assay (formerly known as GEM ExTra) detects single-nucleotide variants, indels, and focal copy number alterations with a mean target coverage of 400× for tumor DNA with 98.8% analytic sensitivity [15]. Despite receiving close to 6 weeks of chemoimmunotherapy, she remained with poor oral intake and was taken for a surgical j-tube placement and diagnostic laparoscopy. A peritoneal biopsy showed metastatic SCC with a PD-L1 CPS of 5 and MMR IHC with absent PMS2. She resumed treatment but developed bleeding from her primary tumor. A palliative resection was attempted but aborted due to the extensive involvement of the vasculature by the primary tumor. A biopsy of the lesser and greater curvature lymph nodes was conducted intraoperatively for further biomarker testing, given the lack of response despite 3 months of therapy by this point. Four nodes from the greater curvature were positive for metastatic carcinoma, one of which consisted of both mucinous adenocarcinoma and SCC morphology within the same LN (Figure 1). The MMR IHC on these samples showed a loss of nuclear expression for PMS2, and PD-L1 testing revealed a CPS of 1. Unfortunately, despite palliative radiotherapy and 2nd-line treatment, her disease continued to progress, and she was referred to hospice and passed away 1 month later.

## 3. Discussion

To our knowledge, there are few cases reported describing intratumor heterogeneity potentially mediating immunotherapy resistance. Our case describes a patient with both gastric adenocarcinoma and SCC of unknown primary site. We postulate that despite different histopathological diagnoses our patient likely had a single metastatic disease process rather than two separate primaries. During the disease course, we sampled the patient’s sites of disease at three timepoints (Figure 2): (1) gastric primary tumor and axillary lymph node, (2) post-treatment peritoneal metastasis, and (3) further post-treatment perigastric lymph nodes. A whole-exome analysis of all the samples shows similar mutations between the metastatic SCC and the primary gastric adenocarcinoma. The excisional LN with contiguous areas of both adenocarcinoma and SCC (Figure 1) was also supportive. For instance, all the tissue samples demonstrated MSI-H as well as somatic mutations mutual to both the gastric adenocarcinoma and the SCC histology seen in the lymph nodes and peritoneal tissue biopsy (Table 1). We pursued more in-depth pairwise comparisons of the SNV profiles for all our samples to enable the testing of clonality or genetic independence between tumor samples (Table 2). The background mutation rate of a detected SNV in at least one of the samples was determined by analyzing the public TCGA pan-cancer database of over 11,000 individual patient samples comprising 66 differing tumor types. Our multiple pairwise test for clonality comparisons indicated that all five samples were significantly associated with a common clonal origin (Table 2). As expected, support for clonality was highest between samples with the same histology, but statistically significant support remained even between samples with differing histology and differing sampling timepoints.

Intrapatient ITH can be a determinant of immunotherapy resistance, which may have been the case in our patient [17]. We suggest that the primary manifestation of heterogeneity is the dedifferentiation of the primary tumor adenocarcinoma into a squamous cell histology in the metastatic sites. Furthermore, within the same perigastric LN, there were tumor regions exhibiting both adenocarcinoma and SCC histologies (Figure 2). A separate primary origin for the SCC was not established, despite evaluation for the usual sites of origin. The metastatic deposition of two primary malignancies into the same lymph node seems less likely to us than the hypothesis that this patient’s metastatic disease was manifested of the same primary tumor with significant clonal evolution and the development of ITH. To further support this premise, patients with MSI-H metastatic GC who have a robust response to pembrolizumab exhibited a reduction in the total size of mutant clones in post-treatment biopsy samples as determined by the ratio of nonsynonymous mutations to synonymous mutations (dN/dS ratio) [17]. In contrast, those with poor responses maintained a high dN/dS ratio, suggesting the evolutionary fitness of mutant subclones [17]. Kwon et al. observed baseline ITH in both responding and non-responding MSI-H patients, but heterogeneity was lost in responding patients, suggesting the maintenance of ITH may be intrinsically resistant to immunotherapy approaches. Interestingly, for our patient, the adenocarcinoma component of the LN biopsies exhibited the lowest number of mutations, presumably representative of some mixed response with the loss of this clone (Table 2). In contrast, the synchronous SCC component of the LN and earlier post-treatment timepoint peritoneal SCC maintained a high number of mutations, indicative of resistance and the overall progressive disease course. 

Furthermore, Wolf et al. studied ITH within melanoma mouse models, demonstrating that the number of distinct clones comprising the tumor and genetic diversity play a role in tumor aggressiveness and its response to ICI [18]. We hypothesize that our patient had a significant amount of ITH, which may have led to immune evasion and a lack of response to ICI. In addition, high ITH can diminish clonal fraction sizes to an extent below a threshold sufficient to mount an immune response despite the presence of neoantigens [19]. We believe that despite being MSI-H in all the tumor samples and regardless of histology, the tumors were likely highly heterogenous, causing the dilution of the neo-antigens [18]. 

We highlight a clinical scenario that suggests the dedifferentiation of the primary tumor led to increased ITH for which we characterized clonality by whole-exome sequencing. We suspect that increased tumor heterogeneity led to the poor response to ICI, despite all tumor samples being MSI-H. We believe further work refining the characterization of intrapatient interlesional heterogeneity can help delineate which patients will derive the most benefit from ICI. Such efforts should be paired with immune gene expression profiling, which can characterize T-cell dysfunction and exclusion within the tumor immune microenvironment [20].

## Figures and Tables

**Figure 1 jcm-11-03413-f001:**
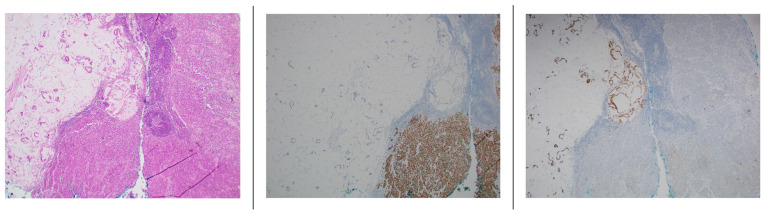
Histologic examination of an excisional lymph node exhibiting contiguous areas of both adenocarcinoma and squamous cell carcinoma. (**Left panel**) representative H&E section. (**Middle panel**) the same section with p40 staining of the squamous cell carcinoma. (**Right panel**) the same section distinguishing areas of CDX2 staining of the adenocarcinoma.

**Figure 2 jcm-11-03413-f002:**
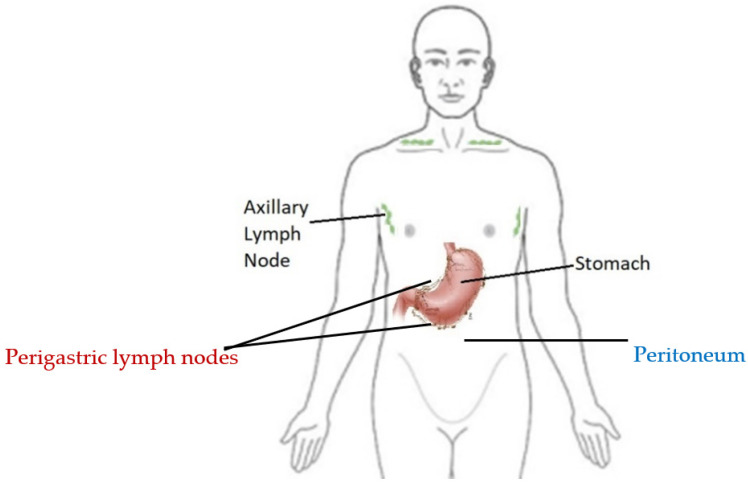
Pre-treatment biopsies of the gastric tumor and right axillary lymph node (black). First post-treatment biopsy of the peritoneum (blue); second post-treatment biopsy of the perigastric lymph nodes (red).

**Table 1 jcm-11-03413-t001:** Actionable and additional alterations of interest detected by Oncomap^TM^ ExTra (Exact Sciences) in the five specimens received for somatic profiling.

Tumor Histology	Adenocarcinoma	SCC	SCC	SCC	Adenocarcinoma
Tissue Source	Stomach	Right Axilla Lymph Node	Peritoneal biopsy	Greater Curvature Lymph Node	Greater Curvature Lymph Node
Timing of Tumor collection relative to start of therapy	15 days prior	11 days prior	41 days post	86 days post	86 days post
**Alterations associated with FDA on/off label therapies**					
ATM:p.R2034*	0.15				
FBXW7:p.E78*	0.17	0.26	0.12	0.07	
MSI-High	Yes	Yes	Yes	Yes	Yes
PMS2:p.E109fs	0.15				
POLD1:p.D1013fs		0.28			
PTCH1:p.S1203fs	0.16		0.15		
TMB-High (≥20 mutations/Mb)	Yes	Yes	Yes	No	No
**Alterations with clinical trial enrollment available**					
ARID1A:p.G240fs	0.14				
ARID1A:p.N1784fs		0.24	0.15	0.06	
ARID2:p.V681fs	0.20	0.37	0.20	0.10	
ASXL1:p.G645fs		0.27			
ATR:p.F1134fs	0.14				
CHEK1:p.T226fs	0.17				
ERCC5:p.K917fs	0.14				
FANCM:p.V1336fs		0.22			
GNAS:p.R844C	0.02				0.03
KRAS:p.G13D		0.38	0.19	0.10	
MLH3:p.N674fs		0.23			
MSH3:p.K383fs		0.24		0.09	
RNF43:p.G659fs	0.17	0.27	0.14		
SLX4:p.P469fs	0.11				
TMB-Intermediate (6–19 mutations/Mb)	No	No	No	Yes	No
TP53:p.D281G	0.02				
**Additional alterations of interest in each tumor type**					
ACVR2A:p.K437fs	0.28			0.12	
ARID1B:p.T71fs				0.05	
B2M:c.68-2A>G			0.12	0.05	
B2M:pL15fs			0.12		
B2M:p.V69fs				0.07	
CLDN18/ARHGAP26 Fusion		Detected			
CREBBP:p.I1084fs	0.16				
JAK1:p.K860fs		0.26			
JAK1:p.P430fs		0.24			
KMT2B:pQ575fs	0.16	0.26		0.08	0.05
KMT2C:p.K2797fs	0.24	0.39			
KMT2C:p.N2842fs		0.18	0.19	0.07	
KMT2C:p.S2237*		0.20	0.23	0.06	
KMT2D:p.A5076fs		0.19	0.12	0.06	
KMT2D:p.Q827fs		0.44	0.27	0.14	
SETDB1:pP451fs	0.20				
TET1:p.K23fs		0.23	0.15		

Table 1 Legend: Table 1 shows the Oncomap^TM^ ExTra report of front-page somatic biomarkers that are either druggable, have clinical trial enrollment, or may be of interest due to prognostic or therapeutic relevance. Mutations are listed with p. annotation unless a splice event. For each sample, if a mutation was present the allelic fraction is listed in corresponding cell. For TMB and MSI, the presence or absence of TMB/MS status was noted as yes or no. Alterations that were present in both gastric and squamous cell carcinoma processes are shaded. No somatic VUS alterations listed.

**Table 2 jcm-11-03413-t002:** Pairwise clonality test of somatic alterations.

Site1_Origin	Site2_Origin	n1	n2	Match	LRstat	maxKSI	LR (*p*-Value)
Stomach_Gastric	RightAxilla_SCC	858	1021	79	448	0.08	<0.01
Stomach_Gastric	Peritoneal_SCC	858	842	71	402	0.08	<0.01
Stomach_Gastric	Lymph_SCC	858	683	59	330	0.08	<0.01
Stomach_Gastric	Lymph_Gastric	858	171	146	1037	0.28	<0.01
Right Axilla_SCC	Peritoneal_SCC	1021	842	683	5900	0.73	<0.01
Right Axilla_SCC	Lymph_SCC	1021	683	567	4776	0.67	<0.01
Right Axilla_SCC	Lymph_Gastric	1021	171	12	51	0.02	<0.01
Peritoneal_SCC	Lymph_SCC	842	683	556	4794	0.73	<0.01
Peritoneal_SCC	Lymph_Gastric	842	171	13	59	0.03	<0.01
Lymph_Gastric	Lymph_SCC	171	683	11	50	0.03	<0.01

Table 2 Legend: In Table 2, each row contains the result of a pairwise comparison of somatic SNV profiles utilizing the SNVtest module of the Clonality R package [16]. Comparisons are ordered by tumor collection dates. Dates are listed in Table 1. n1 = somatic alterations identified for site 1. n2 = somatic alterations identified in tumor site 2. Match = number of matching alterations between two sites. LRStat = likelihood ratio statistics. maxKSI = clonal strength between tumor sites. Note for interpretation: while there is high clonal relatedness between tumor sites with similar origin, especially between the squamous cell carcinomas, as indicated by maxKSi scores of >0.5, predictions suggest clonality between all tumor sites with possibly different backgrounds, as comparisons for clonality hypothesis are *p* < 0.01 across all pairwise comparisons.

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
