# Peer review of "A Case Report of Immunotherapy-Resistant MSI-H Gastric Cancer with Significant Intrapatient Tumoral Heterogeneity Characterized by Histologic Dedifferentiation"

_jcm, 2022, doi:10.3390/jcm11123413_

Round 1

Reviewer 1 Report

Kethireddy and colleagues report a case with both MSI-H gastric cancer and MSI-H SCCs and showed therapeutic response as well as analysis by tumor-only multigene panel. They shows am important message about tumor heterogeneity and therapeutic response. However, they should show if the patient is Lynch syndrome or not by familial history and/or germline DNA mismatch repair gene analysis because all the tumor shows MSI-H

Reviewer 2 Report

The authors presented an advanced stage cancer patient affected by two histologically distinct (but molecularly relevant) tumors, probably as a result of divergent differentiation. Intratumoral heterogeneity is a serious suspect for resistance to immune checkpoint inhibitor therapy and thus the paper is of particular interest in the field of immuno-oncology. The diagnostic and computational methods seem appropriate and adequate, the manuscript is well-prepared, and the discussions sound reasonable.     

Comments:

-       Please include the race/ethnicity of the patient, if ethically permitted.

-       Please add the CPS for the metastatic SCC focus in the axillary lymph node.

-       Please include the full form of all abbreviations the first time they appear in the text (line 148: ITH, and line172: ICI).

-       Intratumoral heterogeneity is proposed as the reason for the resistance to immune checkpoint inhibitor therapy in this patient. I would like to suggest that gene expression profiling or any detected molecular variants may also play a role in this resistance. Did the authors consider this idea? Some computational tools, such as TIDE (Tumor Immune Dysfunction and Exclusion (TIDE) (harvard.edu)), can predict the response to the immune therapy. Did the authors try them? ITH may not be the only reason for ICI resistance in this patient.  

-       Regarding Table 1: 1. Lines 82-83: the authors stated that the “circulating tumor DNA analysis via the Guardant360® assay detected MSI-H and numerous somatic alterations (Table 1). This is contrary to the Table 1 title which says the variants have been detected by OncomapTM ExTra (Exact Sciences); 2. Did the authors mean “The Oncotype MAP™ Pan-Cancer Tissue test” by “OncomapTM ExTra”? 3. Please provide the key specifications for this method (such as the platform and instrument, type of gene enrichment, analytical sensitivity, mean coverage, etc.); 4. Please include the cutoff for TMB; 5. ATM:p.R2034* is written twice. 6. I think RNF43:p.G659fs, PTCH1:p.S1203fs, and KMT2C:p.K2797fs should also get shaded. 7. What does × mean for CLDN18/ARHGAP26 Fusion? 8. Tumor origin is written as “Cancer of Unknown Primary” in column 4, but it is not mentioned like this in the text; 9. Please include the location of LNs in the tissue source row.   

Reviewer 3 Report

An excellent case report using genetic testing between primary gastric  adenocarcinoma and differentiated squamous cell carcinoma in a patient to demonstrate the differentiation and tumor heterogeneity is the cause of treatment resistance. 
